# Publics and UK parliamentarians underestimate the urgency of peaking global greenhouse gas emissions
John Kenny ✉ & Lucas Geese

Intergovernmental Panel on Climate Change assessment reports treat politicians as recipients of information, but not as foci of research efforts. Moreover, academic research on politicians' knowledge concentrates on belief in climate change's anthropogenic cause. Little is known of how aware national parliamentarians are of key findings and policy recommendations from assessment reports. Here, we address this through a survey of 100 Members of Parliament in the United Kingdom on their knowledge of the well-publicised statement from the 6th assessment report of when global greenhouse emissions need to peak for a global temperature increase limit of 1.5 °C to be possible. Parliamentarians overwhelmingly overestimate the time period humanity has left to bend the temperature curve although partisan differences apply. Public surveys in Britain as well as Canada, Chile and Germany show similarly low knowledge, yet being younger, worried about climate change, and having lower levels of conspiracy belief mentality increase accuracy significantly.

The United Nation's Intergovernmental Panel on Climate Change (IPCC) assessment reports are a mammoth task, involving 782 authors, 66,000 peer-reviewed articles and almost 200,000 review comments[1]. They aim to provide an authoritative, objective source of information for policymakers on climate change's causes and effects, as well as to outline pathways for mitigation and adaptation[2]. Their work has been described as "the world's most influential climate report"[3] and "the equivalent of the King James Bible on climate change"[4].

As actors with the capacity to affect meaningful change within their countries and beyond, governments, parliamentarians and other policymakers are the primary targets of dissemination[4]. National governments are intertwined into the process itself to assist in their buy-in. Each government has to grant line-by-line approval to the summary for policymakers until it is agreed by consensus, and they agree to accept the reports as definitive[5].

And yet, surprisingly little is known about the extent to which key messages from their summary reports are comprehended by parliamentarians. This is important given that the reports have been criticised for being overly complex for policymakers or the public to digest[6], including by policymakers themselves[4]. Academics have also sought to raise attention that IPCC reports "treat politicians as a recipient of research findings, not an important object of detailed study"[7]. And yet the wider literature largely does not generally measure politicians' climate change attitudes, including their knowledge of IPCC reports[8]. Understanding whether politicians have knowledge of key findings in IPCC reports is vital given that—despite the political consensus that the reports have—governments regularly follow differing if not contradictory policies to those that would be consistent with the reports' findings[9].

It is only recently that research on climate change and politicians has started to become a focus of peer-reviewed literature[10]. Getting access to politicians is difficult, especially sitting parliamentarians who have strong pressures on their time[11], and thus the lack of research is understandable. However, the lack of awareness of politicians' knowledge of the state of climate science is problematic. For although knowledge does not automatically result in policy action as people may not respond to scientific evidence in a linear way[12–15], it is an important precursor to taking informed judgments. Empirical evidence demonstrates that knowledge can make a difference to politicians' decisions on climate change[16] as well as other policy areas[17].

The limited research that exists on parliamentarians' climate knowledge mostly examines knowledge of climate change's human causes. In anglophone countries, this reveals considerable partisan divides[18–22]. Elsewhere, in a 2010 survey of Peruvian legislators, overall knowledge of ten statements based on the IPCC's 2007 report (AR4) was poor[23]. While important contributions, from such literature we do not know whether parliamentarians internalise how urgent it is to implement deep and rapid decarbonisation. As one study examining knowledge of local politicians in Sweden (among other groups) has demonstrated, knowledge of climate change's causes may be greater than that of its current state and future consequences[24]. Some richer information from MPs has been obtained through interviews[25–27], but these are not designed to be representative.

Tyndall Centre for Climate Change Research, School of Environmental Sciences, University of East Anglia, Norwich, UK. ✉e-mail: john.kenny@uea.ac.uk

Here, we address this gap by examining United Kingdom (UK) Members of Parliament's (MPs') knowledge of a key finding of the IPCC's 6th assessment report: that global greenhouse gas emissions need to peak by 2025 if global average temperature increases are to be kept to 1.5 °C. The question was also asked to a representative sample of the British public. Both surveys were fielded in late 2023. The case and issue selection have particular theoretical and normative value.

On the case, the UK has historically been an international leader on climate change mitigation. Its landmark 2008 Climate Change Act served as the blueprint for many other countries. The UK Parliament was also the first national parliament to declare an environmental and climate emergency in 2019 and it co-hosted the 2021 Glasgow Conference of the Parties[28]. While not negating the political divides that exist on climate policies that have become more prominent in recent years[29,30], it is thus a most likely country for parliamentarians to be aware of the key findings from IPCC reports. If they are not, then that suggests that the dissemination of the messages into other parliaments may also be a problem. For comparative purposes, we draw on public surveys undertaken in Canada, Chile and Germany to examine the individual-level drivers of public responses given to the knowledge question.

On the choice of topic, the "global emissions must peak by 2025 to retain a realistic chance for 1.5 degrees" statement was a key communication of the IPCC report directed at publics and policymakers. As such, it received headline attention in Britain and internationally from across the news spectrum, including the Economist[31], Daily Mail[32], Guardian[33], Euronews[34] and the BBC[35]. Thus, MPs—and publics who pay attention to current affairs —should have been exposed to the finding even in the absence of reading the report. From the perspective of the report lead authors, it was something they emphasised as a key takeaway in media interviews. One lead author referred to the finding as "a bit of a bombshell"[36] while another told the BBC News[35] that:

"I think the report tells us that we've reached the now-or-never point of limiting warming to 1.5 °C. We have to peak our greenhouse gas emissions before 2025 and after that, reduce them very rapidly".

On the United Nations news section, it was also something that one of the co-chairs of Working Group III was quoted on as a takeaway message[37]. Thus, it was both a visible message and one that those involved in the report considered important to highlight, with the BBC News[38] indeed referring to it in a headline as a "key finding".

Moreover, given the limited time period of action, the decisions of these MPs during their period of office (2019–2024) were important to aid in the

collective worldwide effort to achieve the goal. The UK, like many other countries[39], has committed in law to reaching net zero emissions by 2050. Yet if MPs are not aware that urgent action needs to be taken much sooner, then that could be one potential explanatory factor for not meeting long-term climate mitigation commitments at the national level see ref. 40. Our results do demonstrate that knowledge of this particular finding of the IPCC's to be extremely low, with just under 15% of both MPs and publics providing the correct answer of 2025, and over 30% of each stating 2040 or later.

## Results

### Surveys of UK members of parliament

A representative sample of 100 members of the UK House of Commons were polled in September/October 2023 as well as 2002 members of the British public in November/December 2023 (see the "Methods" section for question wording and further details). They were each presented with a one-sentence summary of the IPCC, informed that a 2022 report had provided a timescale by which global greenhouse gases must peak to limit global heating to 1.5 °C above pre-industrial levels, and asked to choose from a choice of options from "2025" to "2050" (in 5 yearly intervals) the timeframe they think this is.

Figure 1 compares top-level responses for both MPs and publics. The distribution is remarkably similar in both groups. Just under 15% gave the correct answer of 2025 with just over 30% opting for 2030. Around a third indicated that global emissions do not need to peak before 2040. The sizeable proportion that answer 2050 may suggest confusion in respondents' minds with the country's Net Zero by 2050 goal. Obviously, respondents might get the right answer simply by guessing, which in aggregate could exaggerate the true number of MPs choosing the correct answer. However, even if we take the 15% of MPs giving the correct answer as an optimistic estimate, this points to a widespread lack of knowledge among both parliamentarians and the British public on the urgency within which action is required, with the vast majority thinking that emissions can peak by the end of the subsequent parliamentary period or later.

Given previous findings that Labour MPs' and voters' belief in the anthropogenic causes of climate change are greater than those of Conservative MPs[18] and voters[41], might there be a similar knowledge gap on the window of action for 1.5 °C? To examine this, in Fig. 2, we breakdown the responses by Conservative and Labour party affiliation for MPs and reported vote at the 2019 General Election for the British public.

**Fig. 1 | British MPs versus public comparison.** This figure provides a breakdown of the percentage of responses provided by MPs (blue bar) and publics (gold bar) across a range of six response options for when global greenhouse gases must peak to limit global heating to 1.5 °C above pre-industrial level.

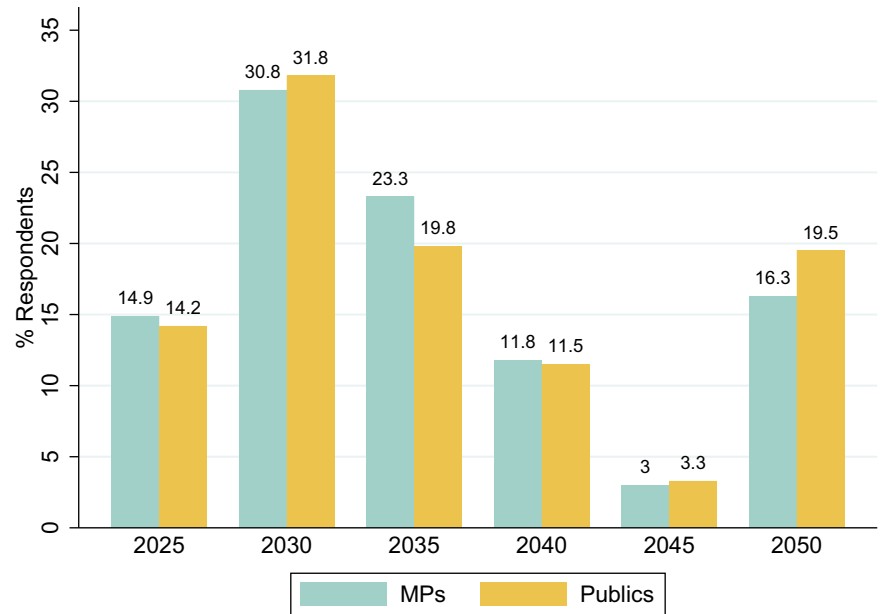

**Fig. 2 | MPs versus publics responses by party.** This figure provides a political party breakdown of the percentage of responses provided by MPs (blue bar) and publics (gold bar) across a range of six response options for when global greenhouse gases must peak to limit global heating to 1.5 °C above pre-industrial level. The top panel provides responses for Conservative MPs and reported Conservative 2019 voters, while the bottom panel provides the responses for Labour MPs and reported Labour 2019 voters.

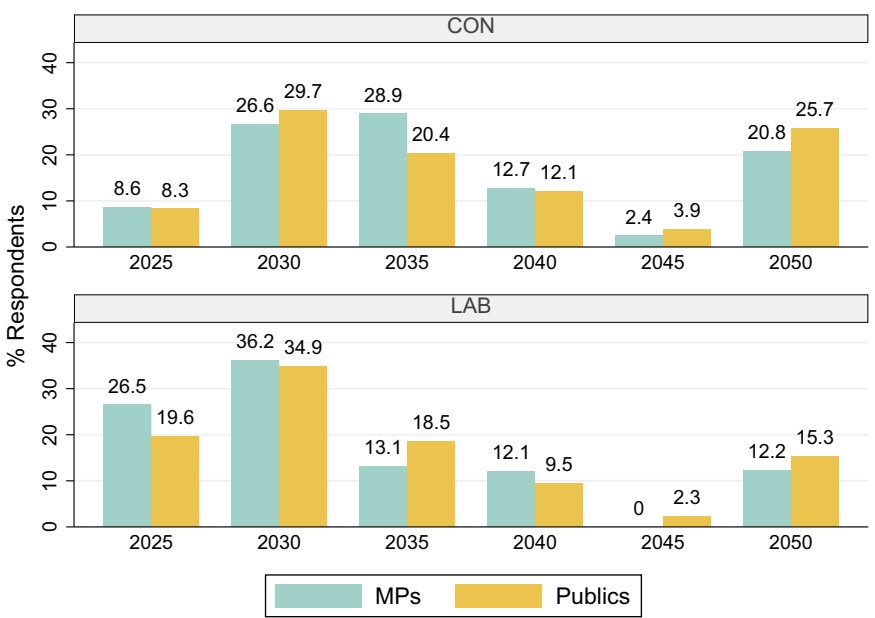

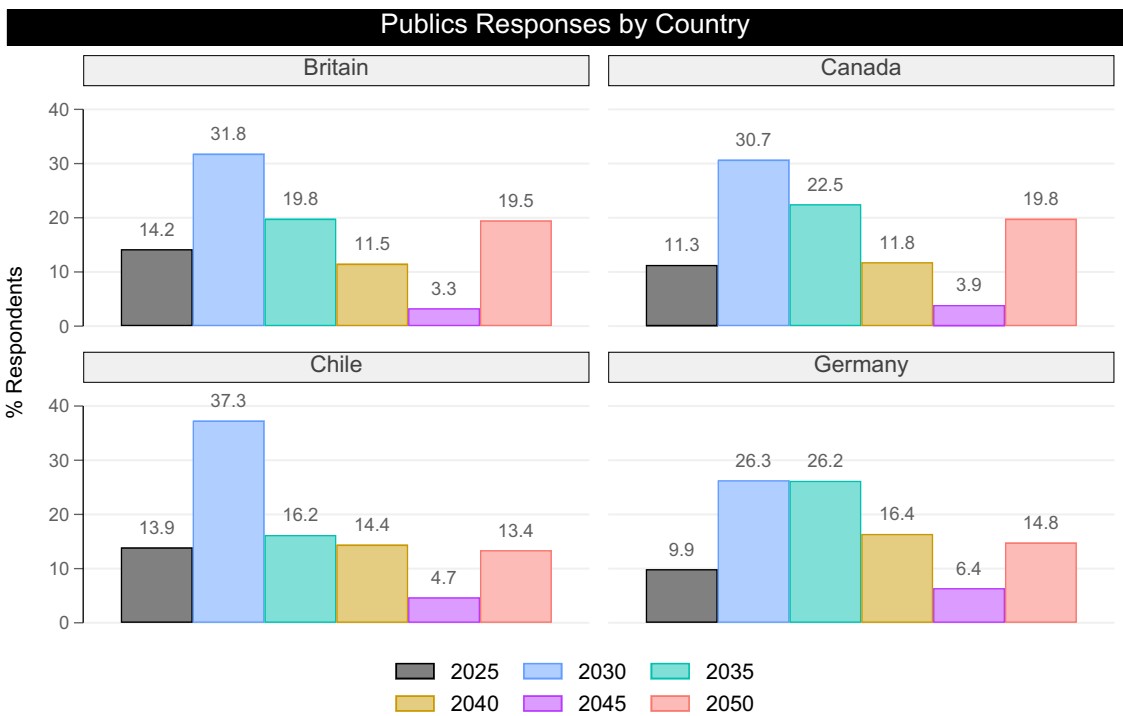

**Fig. 3 | Publics responses by country.** This figure provides a breakdown of the percentage of responses provided by publics in four countries across a range of six response options for when global greenhouse gases must peak to limit global heating to 1.5 °C above pre-industrial level. The top-left panel displays responses in Britain, the top-right panel responses in Canada, the bottom-left panel responses in Chile, and the bottom-right panel responses in Germany. The colour scheme is generated using the Plottig schema[64].

The results show differences between MPs and voters of the same party. 27% of Labour MPs got the answer correct in comparison to 20% of Labour voters, while Labour voters are similar to Labour MPs in thinking the correct answer is 2030 or 2050. There is no difference between Conservative MPs and Conservative voters in those who got the answer correct at just 8.5%. However, 21% of Conservative MPs and 26% of Conservative voters consider that it is the further timeframe of 2050—these are approximately 60 percent higher than the respective answers for Labour MPs and voters. Thus, party affiliation/support is clearly associated with beliefs on the window of opportunity available.

### Surveys of publics in Britain, Canada, Chile and Germany

To assess whether this lack of knowledge is particular to Britain or more widespread, we also asked the question to publics in 3 other countries: Canada, Chile and Germany (See Fig. 3). The response distribution follows a broadly similar pattern in all countries as that observed in Britain, with those selecting the correct answer of 2025 falling to as low as 10% in Germany. Chileans have the highest percentage (37%) of responses selecting 2030, while Germans stand out for being evenly split between 2030 and 2035 as the modal answers. However, in all countries the correct answer "2025" is either the second or third least chosen response.

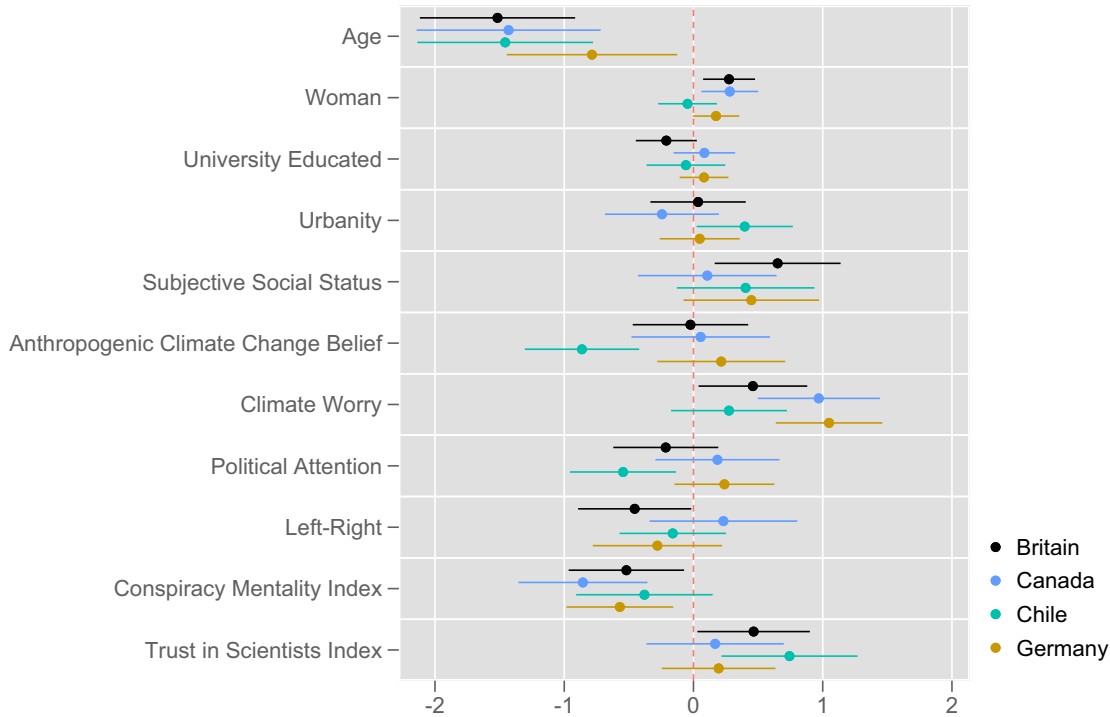

**Fig. 4 | Individual predictors of respondents' degree of accuracy to knowledge question.** This figure provides statistical coefficients (dots) from an ordinary least squares (OLS) regression model regressing responses to the knowledge question on a variety of individual characteristics. Coefficients greater than 0 indicate a positive association, while coefficient less than 0 indicate a negative association. Each coefficient has a 95% confidence interval around it. If the confidence interval overlaps with the vertical line, the association does not surpass the $p < 0.05$ level. If the confidence interval does not overlap with the vertical line, then the association is statistically significant at the $p < 0.05$ level. The colour scheme is generated using the Plottig schema[64].

How do these patterns breakdown for different groups within each country? Given the large numbers of respondents, we can run ordinary least squares multivariate regression models to examine this. We code the dependent variable so that 1 (2050) represents the answer furthest from what the report found, and 6 (2025) represents the correct answer, i.e., higher values represent answers closer to the correct one. In this way, we can account for the degree of accuracy of respondents' answers.

For explanatory variables, we include factors known to correlate with climate change beliefs and attitudes, beginning with sociodemographic predictors, that is age, gender, being university educated, living in an urban or rural area, and a measure of self-assessed social status[41–43]. To this, we add belief in the anthropogenic causes of climate change and worry about it, with greater levels of these having been found to relate to perceptions of the scientific consensus on climate change[44,45]. Moreover, general values, worldviews and political orientation may have substantial predictive power given previous research findings[46]. Therefore, we insert into our models left-right self-placement, a conspiracy belief mentality index, and a trust in scientists index. Given that we are examining knowledge of a report that was heavily publicised in the media, we additionally include self-reported political attention. All variables have been recoded to run from 0 to 1 (see the "Methods" section for further details).

The findings of the models are presented in Fig. 4. The one characteristic that is statistically significant in all countries is age, with younger individuals more likely to be closer to the correct answer. A further two variables are significantly correlated with being closer to the correct response in all countries except Chile: being more worried about climate change and having a lower conspiracy belief mentality. Unique to Chile in attaining statistical significance is residing in a more urban area. In Chile as well as Britain, those who trust scientists more are more likely to be closer to the correct answer. Controlling for all these factors, left-right self-placement only appears as statistically significant in Britain, where the more right-wing are more likely to be incorrect. Adding in vote at the previous election to the

British model as discussed earlier in the paper (model not displayed), we see that this is driven by Conservative voters being further away and Green voters being closer on average to the correct answer than Labour voters.

## Discussion

In this paper, we investigated—firstly—whether MPs in the House of Commons and the British public were aware of a key and widely publicised finding from the IPCC's 6th progress report: that global greenhouse gas emissions need to peak by 2025 to limit global temperature increases to 1.5 °C. Our results demonstrated knowledge of this particular finding to be extremely low, with just under 15% of both MPs and publics providing the correct answer of 2025, and over 30% of each stating 2040 or later. Despite minimal aggregate differences between the MP and publics survey, a key divide was based on political party within MPs and publics, whereby Conservative MPs/voters were less likely to provide the correct answer than Labour MPs/voters. Secondly, we showed that these low levels of knowledge are not confined to publics in Britain but are also recorded by publics in the diverse countries of Canada, Chile and Germany. Greater accuracy among publics was generally predicted by being younger, being worried about climate change, and having lower levels of conspiracy belief mentality.

These results raise important implications for communicating findings from IPCC progress reports. Only a small percentage of MPs in the 2019–2024 House of Commons were aware of the report's important finding that global emissions needed to peak during their period in office for the 1.5 °C target to be achievable. And yet, to act on information, individuals need to be aware of it. That the message is not getting through may be one barrier for the country being off track to meet their 2030 emissions reduction targets, which would require the imposition of more stringent climate mitigation measures as reported in the UK Climate Change Committee's[47] 2024 report. The partisan divides in Britain—whereby MPs of the then ruling Conservative party were less likely to respond correctly than those of the then Labour opposition, and that these divides were mirrored among

their voters—also suggest that whether information is perceived or not and, if so, how it is perceived, may differ according to political mindsets.

The UK is of course not the only country to be off track in meeting its 2030 emissions reduction targets. Taking into account the institutional constraints within countries, a 2024 study suggests that even under the most ambitious scenarios, there is only a 5–45% likelihood of staying below 1.6 °C at peak warming and that without a near-term increase in climate policy ambition that "an even higher overshoot will soon become inevitable"[48]. The other three countries in our publics surveys are all set to miss their 2030 emissions reduction targets as found in independent reports for Canada[49], Chile[50], and Germany[51]. Germany indeed is estimated to have the largest percentage gap within the European Union between its target and what it is set to achieve[51]. Our paper shows that knowledge of the degree of urgency required is low both within the political elite in the UK and in the public across four diverse countries. Politicians pay attention to the views of publics or constituents, whereby concern from below can motivate politicians into decarbonising action, and a lack of concern can act as a barrier[10,52]. Thus, the lack of awareness among members of the public of this key finding from the IPCC report—despite it being widely reported on—is of concern.

This present study has some limitations. While we focus on knowledge of an important policy-relevant finding, we cannot infer whether this low knowledge generalises to knowledge of other findings from the IPCC report. Furthermore, politicians and publics may be in possession of other knowledge and experiences that would propel them to undertake urgent action, and individuals may be aware of the IPCC's findings without believing them or without these facts affecting their viewpoints. For while "scientific knowledge of climate change is clearly imperative for informed policy-making"[53], the mechanism through which knowledge transfers to other facets of climate change beliefs, attitudes and policy support is not always straightforward[15].

As we can identify the knowledge deficit among MPs but not ascertain the reason(s) for this in our study, we call for future research that can investigate such mechanisms through for instance in-depth qualitative interviews with national MPs across the globe on their consumption of IPCC findings and the perceived influence that these do (or do not) have on them. Being able to track MPs' knowledge of a wider variety of findings from IPCC reports would also provide a more nuanced and complete understanding of what facts are landing with the IPCC's core target audience and what is not.

To conclude, in a world of increasing information saturation and disinformation campaigns, getting factual information to permeate through political establishments and the public sphere is by no means an easy task. This is especially the case on a complex challenge like climate change for which certain powerful economic and political interests continue to widely spread misinformation on the facts of the issue[54]. But it is imperative in advance of the 7th progress report that the IPCC and the wider research community can further understand how to effectively get the key scientific messages across to policymakers and the broader public. As time for meaningful action is running out, achieving such a task is of an urgent nature.

## Online methods
### Data collection
**MP survey.** MPs are a notoriously difficult group to contact given severe limits on their time[11]. One approach when carrying out MP surveys is for researchers to initiate contact with them and issue subsequent reminders to complete the survey; however, this typically leads to very low response rates or sample sizes by which it is not possible to make inferential claims on the wider population[55].

To overcome this, our question was placed on an online Omnibus survey of MPs carried out by Savanta, which has a record of obtaining high-quality data with high response rates from MPs[56–58]. Savanta has been running its parliamentary panel for over 20 years. It runs its survey reaching 100 MPs multiple times a year while also engaging with parliamentarians outside fieldwork times to ensure regular participation. During fieldwork,

Savanta monitors quotas and conducts quality control to ensure genuine completion from MPs. However, it is important to note that MPs on the "Government payroll", i.e., Ministers, are not invited because by law these are not allowed to give any opinion that is not the Government policy. Ethics approval for this study—including the procedure for receiving informed consent from participants—was approved on 18th November 2022 by the Faculty of Science Research Ethics Subcommittee at the University of East Anglia (Ethics number: ETH2223-0766).

The field dates were from 6 September to 16 October 2023, with a total sample of 100 MPs. Weights correcting for party affiliation (Conservative, Labour, SNP and Other), region (North, Mid, South), gender (female/male) and age were provided by Savanta and applied in the analysis so that the sample represents the distribution of these characteristics in the House of Commons. As can be seen in Tables 1 and 2 below, the unweighted data closely matched key respondent characteristics in the House of Commons at the time of the survey.

Furthermore, as can be seen in Table 3, the impact of the survey weights on responses given to the knowledge question is minimal.

**Publics survey.** The publics surveys were fielded by Survation, which has a record of accuracy in polling, such as being the only member of the British Polling Council to correctly predict the hung parliament of the 2017 General Election[59,60]. Details of the fieldwork dates, sample size and quotas are included in Table 4. Weights were applied in the analysis so that the sample represents the distribution of key demographic characteristics at the national level; in addition to weighting based on the sampling quotas used, weights in Britain also accounted for votes at the 2019 General Election and the 2016 Brexit referendum; in Canada for votes at the 2021 Federal Election; in Chile for votes at the 2021 Presidential Election; and in Germany for votes at the 2021 Federal Election. Ethics approval for this study—including the procedure for receiving

**Table 1 | Party Breakdown in MP survey compared to House of Commons [unweighted]**

| Party | Respondents | Percentage | Share in House of Commons |
|---|---|---|---|
| Conservative | 51 | 51% | 55% |
| Labour | 32 | 32% | 31% |
| SNP | 6 | 6% | 7% |
| Other | 11 | 11% | 7% |
| Total | **100** | | |

**Table 2 | Gender Breakdown in MP survey compared to House of Commons [unweighted]**

| Party | Respondents | Percentage | Share in House of Commons |
|---|---|---|---|
| Male | 75 | 75% | 66% |
| Female | 25 | 25% | 34% |
| Total | **100** | | |

**Table 3 | Responses to the knowledge question in the MP sample, displaying both the unweighted and unweighted response distributions**

| Answer | Weighted | Unweighted |
|---|---|---|
| 2025 | 14.9% | 16.0% |
| 2030 | 30.8% | 31.0% |
| 2035 | 23.3% | 23.0% |
| 2040 | 11.7% | 10.0% |
| 2045 | 3.0% | 4.0% |
| 2050 | 16.3% | 16.0% |

## Table 4 | Details of public surveys

|  | Fieldwork dates | Sample size | Quotas used | Design effect |
|---|---|---|---|---|
| Britain | 15 November–18 December 2023 | 2002 | -Age<br>-Sex<br>- Geographic region<br>- Education<br>- Household income | 1.23 |
| Canada | 14–30 November 2023 | 1542 | -Age<br>-Sex<br>- Geographic region<br>- Education<br>- Household income | 1.29 |
| Chile | 14 November–4 December 2023 | 1514 | -Age<br>-Sex<br>- Geographic region<br>- Education | 1.41 |
| Germany | 20 November–15 December 2023 | 2177 | -Age<br>-Sex<br>- Geographic region<br>- Education | 1.61 |

informed consent from participants—was approved on 18 November 2022 by the Faculty of Science Research Ethics Subcommittee at the University of East Anglia (Ethics number: ETH2223-0575), and amendments approved by the same subcommittee on 2 October 2023 (Ethics number: ETH2324-0224).

### Survey questions
The wording of the fielded knowledge question is as follows:

"In 2022, the UN IPCC—an international working group of scientists who regularly summarize scientific assessment on climate change—released a series of progress reports.

One of these reports provided a timescale by which global greenhouse gas emissions must peak [i.e., stop increasing] to limit global heating to 1.5 degrees Celsius above pre-industrial levels.

When do you think this is? Even if you are unsure of the answer or are unfamiliar with the reports, please provide your best guess".

Response options were: 2025; 2030; 2035; 2040; 2045; 2050

**Other variables**. *Note: For all questions below, where respondents responded "don't know", these have been treated as missing variables in analyses.*

Age: What is your age; Sex: What is your sex; Education: What is the highest educational level that you have?

Urbanity: Would you describe the place where you live as (1) A farm or town in the country, (2) A country village, (3) A small city or town, (4) The suburbs or outskirts of a big city, (5) A big city.

Subjective social status: In our society, there are groups which tend to be towards the top and groups which tend to be toward the bottom. Below is a scale that runs top to bottom. Where would you put yourself on this scale? (1. Bottom–10. Top).

Past vote at 2019 UK general election: Thinking back to the General Election in December 2019, can you recall which party you voted for in that election?

Left-right: In politics, people sometimes talk of "left" and "right". Where would you place yourself on this scale, where 0 means the left and 10 means the right?

For anthropogenic climate change belief and climate worry, we use the questions as fielded in the European Social Survey[42]. For the conspiracy mentality index, we used the 5-item index of Bruder et al.[61]. The trust in

scientists index is composed of responses to the following two statements, each asked on a 1–5 agree/disagree scale: "I trust scientists to create knowledge that is unbiased and accurate" and "I trust scientists to inform the public on important issues"[62].

### Empirical strategy
For Figs. 1–3, we carried out descriptive crosstabulations and created the bar charts from these. For Fig. 4, we carried out ordinary least squares regression models within each of our four countries, utilising provided survey weights to match demographic targets at the national level. The results are robust to also carrying out ordinal logistic regression models.

For publics, we can exclude respondents who answered the knowledge question in 5 s or less to account for inattentiveness, as well as those who took 1 min or more to respond for potentially looking the answer up. Leaving a sample of 1732 individuals in Britain, this does not significantly change response distributions.

### Reporting summary
Further information on research design is available in the Nature Portfolio Reporting Summary linked to this article.

### Data availability
The data for the publics' surveys have been uploaded to Code Ocean[63]. The data underlying the MP survey is taken directly from crosstabulations provided to the authors by Savanta (https://savanta.com/), the data collector.

### Code availability
The code used for the publics' survey has been uploaded to Code Ocean[63]. The analyses were carried out using Stata 15.1.

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

## Acknowledgements

We thank the editors and the two reviewers for providing helpful suggestions on previous drafts. The work was supported by the European Research Council (via the DeepDCarb Advanced Grant 882601) and the UK Economic and Social Research Council (via the Centre for Climate Change and Social Transformations (CAST) ES/S012257/1).

## Competing interests

The authors declare no competing interests.
