## [Transparent Peer Review file · Communications Earth & Environment]

IPCC's 6th report finding on the urgency of peaking global emissions is underestimated by publics and United Kingdom parliamentarians

Corresponding Author: Dr John Kenny

Version 0:

Decision Letter:

Dear Dr Kenny,

Your manuscript titled "IPCC's key findings on the urgency of climate mitigation are not understood by MPs and publics" has now been seen by 2 reviewers, whose comments are appended below. You will see that they find your work of some potential interest. However, they have raised quite substantial concerns that must be addressed. In light of these comments, we cannot accept the manuscript for publication, but would be interested in considering a revised version that fully addresses these serious concerns.

We hope you will find the reviewers' comments useful as you decide how to proceed. Should additional work allow you to

- address these criticisms (that is, either to incorporate the suggestions or provide a compelling argument why the point made by the reviewer is not valid or relevant to the editorial threshold as outlined below)

AND

- meet our editorial thresholds as outlined below,

then we would be happy to look at a revised manuscript.

In the following, we list our requirements for publication.

*****Provide novel and fully supported insight into the knowledge of parliamentarians and the general public about the findings and recommendations of IPCC reports.**

*****Outline your data and methods in detail, including recruitment method and representativeness of the samples and public survey.**

*****Discuss your findings in detail, reflect on possible limitations, and demonstrate that your data and analysis fully support all your claims.**

If you choose to take up this option, please either highlight all changes in the manuscript text file, or provide a list of the changes to the manuscript with your responses to the reviewers.

When resubmitting, please provide a point-by-point response to the reviewers' comments. Please submit your responses as a separate file, distinct from your cover letter where you can add responses to the Editors' comments that you do not want to be made available to the reviewers. Word files are preferred. We recommend that any figures, tables or graphs that are included in the response to reviewers are also included in the main article or Supplementary Information.

If the revision process takes significantly longer than three months, we will be happy to reconsider your paper at a later date, as long as nothing similar has been accepted for publication at Communications Earth & Environment or published elsewhere in the meantime.

Please use the following link to submit your revised manuscript, point-by-point response to the reviewers' comments with a list of your changes to the manuscript text (which should be in a separate document to any cover letter), a tracked-changes version of the manuscript (as a PDF file) and any completed checklist:

Link Redacted

Please do not hesitate to contact us if you have any questions or would like to discuss the required revisions further. Thank you for the opportunity to review your work.

Best regards,

Miranda Boettcher, PhD
Editorial Board Member
Communications Earth & Environment
orcid.org/0000-0001-7975-4945

Martina Grecequet, PhD
Senior Editor
Communications Earth & Environment

EDITORIAL POLICIES AND FORMAT

If you decide to resubmit your paper, please ensure that your manuscript complies with our editorial policies and complete and upload the checklist below as a Related Manuscript file type with the revised article:

Editorial Policy Policy requirements
(Download the link to your computer as a PDF.)

- Behavioural and social science
- Ecological, evolutionary & environmental sciences
- Life sciences

<https://www.nature.com/documents/nr-reporting-summary.zip>

For your information, you can find some guidance regarding format requirements summarized on the following checklist: (<https://www.nature.com/documents/commsj-phys-style-formatting-checklist-article.pdf>) and formatting guide (<https://www.nature.com/documents/commsj-phys-style-formatting-guide-accept.pdf>).

REVIEWER COMMENTS:

Reviewer #2 (Remarks to the Author):

This manuscript discusses the results of five surveys asking a common multiple-choice question about knowledge of one of the 6th IPCC report conclusions. The results of these surveys suggest that the knowledge of 100 British MPs differs little from the British general public. Moreover, the responses of citizens in the UK, Germany, Canada and Chile are mostly indistinguishable from random guesses.

While I have some more minor questions about generalisability and recruitment, the analysis methods seem appropriate, and the conclusions well justified. The topic is important and the research ambitious, especially in its attempt to measure politician views on this important subject. However, I have two more substantive concerns which make me question the novelty and import of the manuscript's findings.

First, I am not convinced that that the evidence is strong enough to support the authors' conclusions that MP and citizen

knowledge of IPCC findings is “extremely low.” The survey asks respondents to choose a date at which “global greenhouse gas emissions must peak” in order to limit global heating to 1.5 C. While certainly an important conclusion, it is not one that is emphasized in many of the most widely publicized articles or press releases about the IPCC’s report. For instance, the IPCC’s own summary for policymakers does not highlight the 2025 date (https://www.ipcc.ch/report/ar6/syr/downloads/report/IPCC_AR6_SYR_SPM.pdf). Nor did many of the outlets targeted at MP policymakers (e.g., by Chatham House and FT) seem to emphasize this date in most of their reports. I don’t deny that an astute policymaker should be aware of this 2025 date, but the narrowness of the question leads me to doubt that we can make very firm conclusions about how familiar citizens and MPs are with the report’s content. It would have been nice to see the authors attempt to measure multiple dimensions of knowledge. Alternatively, the authors might provide validation that this is the “key finding” that the IPCC authors would hope policymakers to glean. If so, perhaps this null finding represents a failure of communication rather than a failure of politician demand?

Second, I’m not convinced that this analysis represents a significant advance on existing research on climate beliefs and politician/public knowledge. Other studies have likewise concluded that politicians do not differ substantially from citizens in their knowledge of climate change and adaptation (e.g., Pearson et al. 2025) and have attempted to unpack politician demand for climate information (e.g., Pereira et al. 2025). As the manuscript notes, there is also a substantial body of research unpacking public knowledge of climate change (including about the IPCC), often with larger samples and more nuanced survey designs (e.g., Andre et al. 2024). There is a contribution here in examining MP knowledge about the IPCC specifically, but, especially given the narrow scope of the question, I question whether this is a substantial advance on existing work.

As a more minor issue, it would also be helpful to see more detail on the recruitment of politicians. The authors refer to <https://savanta.com/audience-expertise/mps/>, which offers almost little detail on recruitment methods, sample representativeness or attrition. Likewise, and given the use of online panels and survey weights, it is important to know how representative the public surveys are.

Cited

Pearson, Mitya, and Alan Wager. "Not so different: Comparing British MPs' and voters' attitudes to climate change." *Parliamentary Affairs* 78.1 (2025): 53-76.

Pereira, Miguel M., et al. "Encouraging politicians to act on climate: A field experiment with local officials in six countries." *American Journal of Political Science* 69.1 (2025): 148-163.

Andre, Peter, et al. "Globally representative evidence on the actual and perceived support for climate action." *Nature Climate Change* 14.3 (2024): 253-259.

Reviewer #3 (Remarks to the Author):

This paper contributes some evidence for an important knowledge gap – the extent to which politicians and wider publics have understood a key message of the IPCC. It shows conclusively that both groups have not remembered or understood the IPCC’s key message that global emissions need to stabilise by 2025.

This is undoubtedly a significant finding, and for this reason, I support the publication of this paper. However, the reasons for this finding are not clear, and, I believe, are much more complex than the article suggests.

The fact that politicians and publics cannot name the correct year for the emissions peak does not come as a surprise for me. Even as someone who works daily on climate policy, I had to think pretty hard, and estimate that it was 2025 (rather than having ‘remembered’ the fact). I checked with climate-focussed colleagues who said the same. In other words, it is almost certainly possible to be very concerned about climate change, and to be motivated to act, without being able to answer this specific question confidently and correctly.

This is particularly because the IPCC is dealing with global emissions, which are different from national country trajectories. So UK MPs might know that the UK has reduced carbon emissions against a 1990 baseline, and they may well know about the 2050 net zero target for the UK, but they might not know the figure for 2025 emissions peaking in order to reach 1.5. And arguably, unless they are involved in global negotiations, they do not need to, if they know the trajectory for their own legislature.

Having said that, it does not mean that the findings are not worthwhile. In particular, the difference between different political positions is an interesting finding. Again, there could be at least two explanations for this: either, that politicians on the right are less concerned about climate change and so are less inclined to read or retain facts about it; or that, when faced with a question that they do not know the answer to, will pick the one that most represents their own position, ie they are effectively answering the question ‘how urgent is climate action?’ and using the different dates as a proxy for their answer.

More generally, this points to the difficulty of using answers to single survey questions to try to infer positions on wider, complex issues. Ideally, a research approach would either combine different questions (eg a q about climate ‘urgency’) alongside the core question; and/or compare with other topic areas eg health or the economy; and/or combine with

qualitative or free-text data, to be able to more confidently analyse MPs' positions. However, as the authors point out, politicians are notoriously difficult research subjects to access, so the research here may provide the least-worst way of getting some data from a large number of difficult-to-access people.

In conclusion, I think this is useful and significant data, and recommend publication. However, there could be more caveating of the key finding, reflections on limitations and suggestions of what a more comprehensive approach to the research could look like.

Communications Earth & Environment is committed to improving transparency in authorship. As part of our efforts in this direction, we are now requesting that all authors identified as 'corresponding author' create and link their Open Researcher and Contributor Identifier (ORCID) with their account on the Manuscript Tracking System prior to acceptance. ORCID helps the scientific community achieve unambiguous attribution of all scholarly contributions. You can create and link your ORCID from the home page of the Manuscript Tracking System by clicking on 'Modify my Springer Nature account' and following the instructions in the link below. Please also inform all co-authors that they can add their ORCID to their accounts and that they must do so prior to acceptance.

Version 1:

Decision Letter:

Dear Dr Kenny,

Your manuscript titled "Key finding from IPCC on the urgency of climate mitigation is not widely understood by MPs and publics" has now been seen by our reviewers, whose comments appear below. In light of their advice, we are delighted to say that we are happy, in principle, to publish a suitably revised version in Communications Earth & Environment.

We therefore invite you to revise your paper one last time to address the remaining concerns of our reviewer 2. In particular, for publication in Communications Earth & Environment, we request that you (1) remove any references and comments to current politics in the UK, (2) discuss the dependence of your conclusions on weighting assumptions, and (3) provide information on the representativeness of the public survey.

At the same time, we ask that you edit your manuscript to comply with our format requirements and to maximise the accessibility and therefore the impact of your work.

EDITORIAL REQUESTS:

****Please take care to match our formatting and policy requirements. We will check revised manuscript and return manuscripts that do not comply. Such requests will lead to delays. ****

SUBMISSION INFORMATION:

OPEN ACCESS:

Communications Earth & Environment is a fully open access journal. Articles are made freely accessible on publication. For further information about article processing charges, open access funding, and advice and support from Nature Portfolio, please visit <https://www.nature.com/commsenv/open-access>

Link Redacted

Best regards,

Miranda Böttcher, PhD
Editorial Board Member
Communications Earth & Environment

Martina Grecequet, PhD
Senior Editor,
Communications Earth & Environment

REVIEWERS' COMMENTS:

Reviewer #2 (Remarks to the Author):

I still find the contribution of this manuscript to be somewhat narrow given the limited scope of the survey questions and the extent of existing literature on elite knowledge; however the authors have done a much better job of explaining their contribution and not over-stating their conclusions. I'm happy to support moving forward if the editors see this contribution as sufficient to publication.

I don't require further review, but there are a couple minor things I would suggest revising before publication:

1. To my mind, the commentary on current politics in the UK does not belong in a scientific article. While it's reasonable to claim that changing policies in the UK are partly the result of ignorance, the data in the manuscript do not support that conclusion, especially given the fact that knowledge of the IPCC findings in the UK are not particularly unusual. Stick to what the data say.
2. The manuscript should provide analysis with and without sample weights to assess dependence of the conclusions to the weighting assumptions.
3. Unless I missed it somewhere, the revised manuscript still does provide the requested information on the representativeness of the public surveys.

Reviewer #3 (Remarks to the Author):

I have read the revisions made to the manuscript and am satisfied that they address the points raised in my initial review.

** Visit Nature Portfolio's author and referees' website at <http://www.nature.com/authors> for information about policies, services and author benefits**

Many thanks for this valuable feedback and for providing us with the opportunity to revise our manuscript for further consideration. In light of the very helpful reviews and feedback, we have undertaken revisions to the manuscript to address the issues raised. Below, we address each point raised and detail how we have adapted the manuscript in response. The reviewers' comments are in black, while our responses are in red text. When page numbers are referenced, in all cases these refer to the document with the changes tracked rather than the clean version.

REVIEWER COMMENTS:

Reviewer #2 (Remarks to the Author):

2.1 This manuscript discusses the results of five surveys asking a common multiple-choice question about knowledge of one of the 6th IPCC report conclusions. The results of these surveys suggest that the knowledge of 100 British MPs differs little from the British general public. Moreover, the responses of citizens in the UK, Germany, Canada and Chile are mostly indistinguishable from random guesses.

While I have some more minor questions about generalisability and recruitment, the analysis methods seem appropriate, and the conclusions well justified. The topic is important and the research ambitious, especially in its attempt to measure politician views on this important subject.

Thank you

2.2 However, I have two more substantive concerns which make me question the novelty and import of the manuscript's findings.

First, I am not convinced that that the evidence is strong enough to support the authors' conclusions that MP and citizen knowledge of IPCC findings is "extremely low." The survey asks respondents to choose a date at which "global greenhouse gas emissions must peak" in order to limit global heating to 1.5 C. While certainly an important conclusion, it is not one that is emphasized in many of the most widely publicized articles or press releases about the IPCC's report. For instance, the IPCC's own summary for policymakers does not highlight the 2025 date (https://www.ipcc.ch/report/ar6/syr/downloads/report/IPCC_AR6_SYR_SPM.pdf). Nor did many of the outlets targeted at MP policymakers (e.g., by Chatham House and FT) seem to emphasize this date in most of their reports. I don't deny that an astute policymaker should be aware of this 2025 date, but the narrowness of the question leads me to doubt that we can make very firm conclusions about how familiar citizens and MPs are with the report's content. It would have been nice to see the authors attempt to measure multiple dimensions of knowledge. Alternatively, the authors might provide validation that this is the "key finding"

that the IPCC authors would hope policymakers to glean. If so, perhaps this null finding represents a failure of communication rather than a failure of politician demand?

We agree that we do not want to overclaim or overgeneralise from our results. In our revised manuscript, we are thus more careful that our results are not drawn upon to make inferences to the IPCC findings as a whole. We have revised our title from “IPCC’s key findings on the urgency of climate change mitigation are not understood by MPs and publics” to “Key finding from IPCC on the urgency of climate mitigation is *not widely understood* by MPs and publics”. This thus better reflects our data which, as noted, focuses on one finding rather than a broader suite of findings.

In this latest revision, we also provide more evidence for this fact being emphasised in widely publicised articles about the report as an important takeaway message as suggested. Importantly, we now include quotes given by authors of the report itself, who chose to focus on this aspect in their media appearances as what they perceived as a vital takeaway point from the report as follows on pg 4/5:

“From the perspective of the report lead authors, it was something they emphasised as a key takeaway in media interviews. Julia Steinberger referred to the finding as ‘a bit of a bombshell’³⁶ while Heleen De Coninck told the BBC News³⁵ that:

‘I think the report tells us that we’ve reached the now-or-never point of limiting warming to 1.5C. We have to peak our greenhouse gas emissions before 2025 and after that, reduce them very rapidly’

On the United Nations news section, it was also something the co-chair of Working Group III Jim Skea was quoted on as a takeaway message³⁷. Thus, it was both a visible message and one that those involved in the report considered important to highlight, with the BBC News³⁸ indeed referring to it in a headline as a ‘key finding.’”

On consumption of news sources by MPs, this finding was covered on multiple occasions by the BBC, which was listed as the most frequent source of news for MPs during this terms of office¹ as well as the most consumed news source by the general public². Thus, it is one that should have been visible to both MPs and the public.

Moreover, on outlets that are more targeted at policymakers than the general public, as listed in the previous submission *The Economist* – widely read by MPs and indeed emphasised by the House of Commons Library as one of the sources available to MPs through their service³ – led with this in their headline. While we had not listed the headline in full – just providing a link to the article – it read as follows: “The latest IPCC report argues that stabilising the climate will require fast action. Emissions must peak by 2025 for the world to have a chance of meeting

¹ <https://savanta.com/knowledge-centre/view/where-mps-get-their-news/>

² <https://www.ofcom.org.uk/siteassets/resources/documents/research-and-data/tv-radio-and-on-demand-research/tv-research/news/news-consumption-2024/news-consumption-in-the-uk-2024-report.pdf?v=379621>

³ <https://commonslibrary.parliament.uk/content/uploads/2019/12/Commons-Library-Information-Guide.pdf>

the Paris goals". What is also important about this particular finding is that it was covered by both left-leaning news sources (e.g. the Guardian) and right-leaning news sources (e.g. The Daily Mail).

We hope this extra information provides some assurances that it was a key finding that IPCC authors hoped policymakers to glean, and that it was one that there was a reasonable chance that MPs in our sample would have been exposed to it given its coverage in outlets that they consume. While we would also have liked to test knowledge of other parts of the report, this was not possible as part of this project due to the space constraints on this elite survey. We now list this as a limitation to the study in the conclusion section (pg. 13), and recommend further research in this area.

2.3 Second, I'm not convinced that this analysis represents a significant advance on existing research on climate beliefs and politician/public knowledge. Other studies have likewise concluded that politicians do not differ substantially from citizens in their knowledge of climate change and adaptation (e.g., Pearson et al. 2025) and have attempted to unpack politician demand for climate information (e.g., Pereira et al. 2025). As the manuscript notes, there is also a substantial body of research unpacking public knowledge of climate change (including about the IPCC), often with larger samples and more nuanced survey designs (e.g., Andre et al. 2024). There is a contribution here in examining MP knowledge about the IPCC specifically, but, especially given the narrow scope of the question, I question whether this is a substantial advance on existing work.

Thank you for highlighting these studies to us. They all make valuable contributions to the literature and are worthwhile engaging with. We had already referenced the Pearson and Wager article, and we have now incorporated a reference to the Pereira et al paper.

We believe that our contribution does make a significant advance on existing research. A primary contribution is in examining MP knowledge about the IPCC reports specifically, and we hope that – in our response to your previous point – we have now provided further evidence on the worthiness of collection responses to our chosen questions.

Notwithstanding this, the study also makes advances on the above literature mentioned. Following the recent framework of Kenny et al (2024), we recognise that questions used in survey research on climate change elicit many different components that are not interchangeable. Our research focus on the "knowledge" component. Pearson and Wager are primarily concerned in their study with attitudes towards climate policy support and policy preferences, or the beliefs about the societal effects of particular policies which are very different. The one question they ask that centres around knowledge/beliefs of the current state of affairs is with regard to the anthropogenic causes of climate change - which as we note in our manuscript referencing other studies is the one climate knowledge question that has been most widely asked about in existing MP surveys - and for which they do observe substantial differences between MPs' and publics' views. Yet the phrasing of their question in

terms of “which of these comes closest to your view” captures more of a belief than a knowledge dimension. For as Fischer and van den Broek (2021: 117)⁴ remark, ‘Where climate change knowledge reflects the accuracy of people’s understanding of climate change information, climate change beliefs are not necessarily evaluated for their correctness, but reflect a judgment in relation to climate change’. Our contribution is on the knowledge rather than beliefs.

The Periera et al study also has a different focus to ours. Firstly, in their study information is used as the *treatment* rather than as the object of study. While it makes a valuable contribution to the literature, our study takes a step back from this by examining the knowledge of information rather than the effect that information has on policymakers beliefs. Secondly, their focus is on local politicians rather than national politicians which – as the recent systematic review by Moore et al (2024⁵) demonstrates – face different challenges, and in particular would not be expected to be as knowledgeable as national politicians on average given resource constraints and lower levels of professionalisation. National legislatures are also those with the most power to implement changes which makes studying their knowledge particularly important.

We do recognise that there are a number of existing studies that unpack public knowledge of climate change, including in relation to knowledge of the IPCC [taking note in particular of the aforementioned review chapter by Fischer and van den Broek, and the review article from Kenny et al (2024)⁶]. The particular advance that we bring with our publics data is by presenting this in tandem with the findings from the UK MP data.

In sum, we believe that our manuscript makes a substantial advance to the existing literature based on its focus on knowledge of MP’s knowledge of a key finding from an IPCC report, and the design of our study which offers a perspective that is not currently present in existing studies.

2.4 As a more minor issue, it would also be helpful to see more detail on the recruitment of politicians. The authors refer to <https://savanta.com/audience-expertise/mps/>, which offers almost little detail on recruitment methods, sample representativeness or attrition. Likewise, and given the use of online panels and survey weights, it is important to know how representative the public surveys are.

⁴ Fischer, H, and K Van den Broek. 2021. “Climate Change Knowledge, Meta-Knowledge and Beliefs.” In *Research Handbook on Environmental Sociology*, edited by Axel Franzen and Sebastian Mader, 116–32. Cheltenham: Edward Elgar Publishing.

⁵ Moore, B. et al. 2024. Politicians and climate change: A systematic review of the literature. *WIREs Climate Change* 15

⁶ Kenny, J. et al. 2024. A framework for classifying climate change questions used in public opinion surveys. *Environmental Politics*

We are very happy to provide this information. We have added more information on Savanta's recruitment methods on pg. 16 in the methods section. On the sample representativeness, we now provide in Tables 1 and 2 of the methods section (pg. 16/17) a breakdown of the unweighted characteristics of our sample according to party affiliation and gender in our sample in comparison to the characteristics of these groups in the House of Commons at the time of the survey, and show that these compare very favourably.

On the publics surveys, we realise we had not previously provided information of the quotas we employed. These are now included in Table 3 in the online methods section (pg. 17). We now also add the design effect statistic⁷ in this table as an indicator of the quality of the data.

Reviewer #3 (Remarks to the Author):

3.1 This paper contributes some evidence for an important knowledge gap – the extent to which politicians and wider publics have understood a key message of the IPCC. It shows conclusively that both groups have not remembered or understood the IPCC's key message that global emissions need to stabilise by 2025.

This is undoubtedly a significant finding, and for this reason, I support the publication of this paper. However, the reasons for this finding are not clear, and, I believe, are much more complex than the article suggests.

Thank you. We agree that we needed to add more detail and caveats with regards to the reasons, and have now added this in (as outlined in our responses below)

3.2 The fact that politicians and publics cannot name the correct year for the emissions peak does not come as a surprise for me. Even as someone who works daily on climate policy, I had to think pretty hard, and estimate that it was 2025 (rather than having 'remembered' the fact). I checked with climate-focussed colleagues who said the same. In other words, it is almost certainly possible to be very concerned about climate change, and to be motivated to act, without being able to answer this specific question confidently and correctly.

We agree with this assessment and had not wished to convey that this knowledge was a proxy for concern or motivation to act. As we had caveated in the introduction on pg. 3 "For although knowledge does not automatically result in policy action as people may not respond to scientific evidence in a linear way, it is an important precursor to taking informed judgments." We had not repeated similar caveats in the discussion which was an oversight on our part. We have now added in the following on pg. 13/14 so that readers do not get this impression:

⁷ See <https://methods.sagepub.com/reference/encyclopedia-of-survey-research-methods/n133.xml>

“While we focus on knowledge of an important policy-relevant finding, we cannot infer whether this low knowledge generalises to knowledge of other findings from the IPCC report. Furthermore, politicians and publics may be in possession of other knowledge and experiences that would propel them to undertake urgent action, and individuals may be aware of the IPCC’s findings without believing them or these facts affecting their viewpoints. For while ‘scientific knowledge of climate change is clearly imperative for informed policy-making’, the mechanism through which knowledge transfers to other facets of climate change beliefs, attitudes and policy support is not always straightforward. “

3.3 This is particularly because the IPCC is dealing with global emissions, which are different from national country trajectories. So UK MPs might know that the UK has reduced carbon emissions against a 1990 baseline, and they may well know about the 2050 net zero target for the UK, but they might not know the figure for 2025 emissions peaking in order to reach 1.5. And arguably, unless they are involved in global negotiations, they do not need to, if they know the trajectory for their own legislature.

This is very true. One would expect knowledge about the 2050 net zero target to be much higher. Yet – since our first submission – we have seen an increase in politicians and parties in the UK rollback on the necessity of even aiming for net zero by 2050, which at the very least suggests a mismatch between the prescribed policy goals of many MPs and the recommendations of the IPCC. To further relate our findings to contemporary developments, we have added the following in the discussion section (pg. 14):

“However, the results of our research can be taken in tandem with other developments. In qualitative interviews with UK political actors, a recent study found the presence of a growing narrative among substantial parts of the political elite that not only is climate change mitigation not considered urgent, but even too expensive to consider feasible pursuing²⁷. The previous political consensus on reaching net zero by 2050 when it was introduced in parliament in 2019 has also recently broken down. The Conservative Party abandoned the policy in 2025 and the populist-right party Reform UK are vocally against the objective of pursuing net zero at all, in contrast to support among the other parties to reach this target by then or earlier⁵⁴.

Such divides that have recently spilled into the open echo the Labour-Conservative MP differences in knowledge of the IPCC finding investigated in our study. They also raise questions as to whether a lack of knowledge about the urgency of the problem may be somewhat responsible for these recent calls to reduce rather than ramp up climate action, or whether scepticism towards the scientific findings or other ideological perspectives are the primary drivers.”

3.4 Having said that, it does not mean that the findings are not worthwhile. In particular, the difference between different political positions is an interesting finding. Again, there could be at least two explanations for this: either, that politicians on the right are less concerned about climate change and so are less inclined to read or retain facts about it; or that, when faced with a question that they do not know the answer to, will pick the one that most represents their own position, ie they are effectively answering the question ‘how urgent is climate action?’ and using the different dates as a proxy for their answer.

More generally, this points to the difficulty of using answers to single survey questions to try to infer positions on wider, complex issues. Ideally, a research approach would either combine different questions (eg a q about climate ‘urgency’) alongside the core question; and/or

compare with other topic areas eg health or the economy; and/or combine with qualitative or free-text data, to be able to more confidently analyse MPs' positions. However, as the authors point out, politicians are notoriously difficult research subjects to access, so the research here may provide the least-worst way of getting some data from a large number of difficult-to-access people.

This point is very well taken. We have added the following text to our discussion on pg. 14 to reflect how our findings could be built upon in future research:

“As we can identify the knowledge deficit among MPs but not ascertain the reason(s) for this in our study, we call for future research that can investigate such mechanisms through for instance in-depth qualitative interviews with national MPs across the globe on their consumption of IPCC findings and the perceived influence that these do (or do not) have on them. Being able to track MPs' knowledge of a wider variety of findings from IPCC reports would also provide a more nuanced and complete understanding of what facts are landing with the IPCC's core target audience and what is not.”

3.5 In conclusion, I think this is useful and significant data, and recommend publication. However, there could be more caveating of the key finding, reflections on limitations and suggestions of what a more comprehensive approach to the research could look like.

Thank you. We have endeavoured to provide such reflections in this revised version.

Many thanks for this valuable feedback and for providing us with the opportunity to revise our manuscript for further consideration. In light of the very helpful reviews and feedback, we have undertaken revisions to the manuscript to address the issues raised. Below, we address each point raised and detail how we have adapted the manuscript in response. The reviewers' comments are in black, while our responses are in red text.

REVIEWERS' COMMENTS:

Reviewer #2 (Remarks to the Author):

I still find the contribution of this manuscript to be somewhat narrow given the limited scope of the survey questions and the extent of existing literature on elite knowledge; however the authors have done a much better job of explaining their contribution and not over-stating their conclusions. I'm happy to support moving forward if the editors see this contribution as sufficient to publication.

We are glad that the reviewer considers the manuscript to have improved and are thankful for their feedback.

I don't require further review, but there are a couple minor things I would suggest revising before publication:

1. To my mind, the commentary on current politics in the UK does not belong in a scientific article. While it's reasonable to claim that changing policies in the UK are partly the result of ignorance, the data in the manuscript do not support that conclusion, especially given the fact that knowledge of the IPCC findings in the UK are not particularly unusual. Stick to what the data say.

We have now removed the two paragraphs in the discussion section that we had added as part of the last revision in which we related our findings to contemporary political developments.

2. The manuscript should provide analysis with and without sample weights to assess dependence of the conclusions to the weighting assumptions.

In the new Table 3 in the online methods section, we now display the distribution of responses among MPs to the knowledge question both with and without the inclusion of weights and demonstrate that the findings are robust to doing so.

3. Unless I missed it somewhere, the revised manuscript still does provide the requested information on the representativeness of the public surveys.

We did provide this in the previous revision. In Table 4, we included the design effect statistic for each country, and in the methods section we also provided more detailed information on the variables that were used as part of our quota sampling.